# Targeting Programmed Cell Death in Flap Ischemia/Reperfusion Injury

**DOI:** 10.3390/biom15070911

**Published:** 2025-06-20

**Authors:** Shengyue Liu, Xiaohe Xiong, Lei Chen, Jiaqi Hu, Ping Luo, Zhanpeng Ou, Fugui Zhang

**Affiliations:** 1Department of Oral and Maxillofacial Surgery, The Affiliated Stomatological Hospital of Chongqing Medical University, Chongqing 401147, China; 2024121470@stu.cqmu.edu.cn; 2Chongqing Municipal Key Laboratory of Oral Biomedical Engineering of Higher Education, The Affiliated Stomatological Hospital of Chongqing Medical University, Chongqing 401147, China; 2022120697@stu.cqmu.edu.cn (X.X.); 2023121365@stu.cqmu.edu.cn (L.C.); 2023121379@stu.cqmu.edu.cn (J.H.); 2023121395@stu.cqmu.edu.cn (P.L.); 3Chongqing Key Laboratory of Oral Diseases, The Affiliated Stomatological Hospital of Chongqing Medical University, Chongqing 401147, China

**Keywords:** skin flap necrosis, ischemia–reperfusion injury, programmed cell death, targeted therapy

## Abstract

A skin flap is a composite tissue unit comprising skin and subcutaneous fat with an intact vascular supply. Skin flaps are commonly employed for wound reconstruction, transplantation of damaged tissues, and cosmetic procedures. However, flap necrosis resulting from ischemia/reperfusion injury (IRI) is a frequent complication, leading to surgical failure. Therefore, This review systematically summarizes the mechanisms and therapeutic interventions targeting specific modalities of programmed cell death (PCD) in the context of IRI compromising flap survival. These interventions encompass a range of strategies, including preconditioning, systemic administration, and local drug delivery. Furthermore, we summarize key therapeutic targets for various types of PCD, along with shared pathways and therapies applicable across multiple PCD modalities. The findings presented in this review validate the feasibility of targeted therapies against PCD to prevent post-reconstructive flap necrosis. These findings provide novel strategies, such as targeting common pathways in PCD and leveraging diverse biomaterials, to enhance therapeutic outcomes. Further clinical investigations are warranted to target PCD pathways for the treatment of flap necrosis.

## 1. Introduction

Flaps are widely used in reconstruction and plastic surgery to address tissue defects. However, post-surgical necrosis of flaps caused by ischemia–reperfusion injury (IRI) remains unresolved. IRI denotes the paradoxical exacerbation of cellular dysfunction and death following the restoration of blood flow to previously ischemic tissues, representing a principal etiology of flap necrosis (Figure 1). Specifically, during tissue ischemia, the deprivation of oxygen and nutrients precipitates cellular dysfunction and activates a multitude of cell death mechanisms; however, upon reperfusion, the resurgence of oxidative stress, free radical production, and inflammatory responses in the local tissue milieu exacerbates tissue damage and cell death. Following ischemia/reperfusion (I/R), the generation and accumulation of reactive oxygen species (ROS) are amplified. This cascade subsequently induces apoptosis, compromises cellular membrane integrity, and triggers inflammatory responses, ultimately culminating in flap necrosis [1]. Recent advancements in addressing flap necrosis have focused on several key strategies, including anti-inflammatory therapies, stem cell-based interventions, and antioxidant treatments [2,3].

Among the mechanisms causing necrosis of flaps, the role of programmed cell death (PCD) is indispensable. PCD refers to a highly regulated process of cellular demise executed through intrinsic pathways under either physiological or pathological conditions. Recent investigations have categorized PCD into several distinct modalities, including apoptosis, necroptosis, pyroptosis, ferroptosis, PANoptosis, and cuproptosis, among others [4,5]. However, due to the complex molecular mechanisms underlying various forms of PCD, current research has yet to elucidate the regulatory network governing the interplay among different PCD pathways. This review summarizes current research and therapeutic approaches related to flap necrosis associated with cell death, elucidates the mechanisms of flap necrosis, and provides insights for future research and treatment.

## 2. Methodology

We conducted a narrative review of studies published from October 2003 to January 2025, sourced from the PubMed, Web of Science, and Scopus databases. After excluding 490 abstracts unrelated to the topic, 70 non-English articles, 333 duplicates, and 32 studies with irrelevant content or poor quality upon full-text evaluation, a total of 204 articles were included in the final analysis.

## 3. Signaling Pathways and Regulatory Networks of Programmed Cell Death

We first briefly summarize the basic concepts of different types of PCD that included in this review (Figure 2). Apoptosis, a rigorously regulated form of programmed cell death, is essential for the maintenance of homeostasis within multicellular organisms. Orchestrated by a complicated molecular signaling network, apoptosis is typically triggered in response to cellular stress or damage, functioning to eliminate damaged or unnecessary cells while avoiding an inflammatory response. The core mechanisms governing apoptosis encompass both extrinsic and intrinsic pathways. The extrinsic pathway is initiated by death receptors. Upon ligand binding, such as TNF-α or FasL, death receptors assemble the death-inducing signaling complex (DISC) on the cytoplasmic side. This subsequently activates caspases containing a death effector domain (DED), such as Caspase-8, which directly cleaves effector caspases, including Caspase-3, thereby inducing apoptosis [6]. The intrinsic mitochondrial pathway is triggered by DNA damage, ROS, or I/R stress, which activates BH3-only proteins, leading to BAX/BAK activation. This results in mitochondrial outer-membrane permeabilization (MOMP), cytochrome c release and assembly with APAF-1 to form apoptotic bodies, activating Caspase-9, and finally activating Caspase-3 and Caspase-7 to cause cell death [7].

Autophagy is a highly conserved cellular degradation process that primarily involves the formation of double-membrane autophagosomes, which transport damaged organelles, proteins, and other cytoplasmic components to lysosomes for degradation, thereby maintaining cellular homeostasis and responding to various stress stimuli. Autophagy is regulated by numerous signaling pathways, with the AMPK–mTOR–ULK1 axis serving as a central regulatory hub for initiating this process. Under conditions of cellular stress, such as ATP depletion and elevated ROS levels, AMPK is activated, leading to the phosphorylation of ULK1 at Ser317/Ser777, thereby triggering autophagy. In contrast, under nutrient-rich conditions, mTORC1 functions as a negative regulator, suppressing autophagy initiation by inhibiting the binding of phosphorylated ULK1 at Ser757 to AMPK [8]. Furthermore, multiple signaling pathways, including p38–MAPK, are also involved in the induction of autophagy [9].

Pyroptosis represents another form of programmed cell death, primarily instigated by the activation of inflammasomes, particularly the NLRP3 inflammasome [10]. During IRI, the NLRP3 inflammasome is activated, leading to the cleavage of Caspase-1 and subsequent processing of gasdermin D. This cascade precipitates pore formation on cellular membrane, causing cell rupture and lysis, followed by the release of pro-inflammatory cytokines, including IL-1β and IL-18, thereby initiating an inflammatory response and more cell death [11].

Necroptosis, a regulated form of cell death characterized by necrotic hallmarks, is orchestrated by intracellular signaling cascades and specific molecular mediators. Unlike classical apoptosis, necroptosis occurs independently of caspase activation. Instead, it relies on receptor-interacting protein (RIP) kinases and their downstream effector molecules, such as Mixed Lineage Kinase Domain-Like Protein (MLKL), to mediate cell death. This process is primarily regulated by a signaling cascade comprising RIP1, RIP3, and MLKL. Upon stimulation by inflammatory cytokines, such as TNF-α, RIP1 is recruited and activated. Following the formation of a complex between RIP1 and RIP3, MLKL is subsequently phosphorylated and activated, leading to membrane rupture and subsequently the leakage of cellular contents [12,13].

Ferroptosis is an iron-dependent, lipid-peroxidation-driven mechanism of cell death. The key hallmarks of this process include the aberrant accumulation of intracellular iron, the generation of ROS, and the peroxidation of polyunsaturated fatty acids (PUFAs) on cellular membranes [14]. Unlike apoptosis or necroptosis, ferroptosis is not solely dependent on the dysregulation of iron metabolism but also involves the impact of oxidative stress. Iron promotes the generation of ROS via the Fenton reaction, thereby inducing significant oxidative damage to lipid membranes, and resulting in the accumulation of lipid peroxides [15]. These peroxides disrupt the integrity of the cellular membrane, ultimately leading to cell death.

PANoptosis is a recently identified mode of PCD, distinguished by the intricate interplay of apoptosis, necroptosis, and pyroptosis pathways. The activation of downstream cell death pathways is mediated by the formation of a PANoptosome complex, a process implicated in the pathogenesis of various diseases. Within the mechanisms regulating PANoptosis, inflammasomes play a pivotal role [16]. Inflammasomes are critical sensor complexes within the innate immune system that detect pathogen-associated molecular patterns (PAMPs) and damage-associated molecular patterns (DAMPs). Upon activation, these complexes trigger Caspase-1 activation, subsequently promoting the release of pro-inflammatory cytokines, including IL-1β and IL-18, and inducing pyroptosis [17]. Different inflammasome subtypes, including NLRP3, AIM2, and RIPK1, contribute to the assembly of distinct PANoptosome complexes, thereby playing a critical role in the regulation of PANoptosis [17].

Current research in the fields of renal injury, tumor, and ischemic stroke has confirmed the feasibility of targeting PCD as a therapeutic strategy [11,18,19,20]. Concurrently, in studies addressing flap necrosis, an increasing number of researchers have also begun to take targeted PCD as the research focus and have preliminarily explored the important role of PCD-targeted treatment of flap necrosis. This review systematically summarizes these studies, providing a theoretical foundation for targeted interventions in flap necrosis.

## 4. Mechanisms and Therapeutic Strategies of Programmed Cell Death in Flap Necrosis

### 4.1. Mechanisms of Apoptotic in Flap Necrosis and Targeted Therapeutic Strategies

We further investigate the mechanisms of different types of PCD that relate to flap necrosis and the potential therapeutic strategies (Figure 2, Table 1). Apoptosis is one of the earliest-identified types of cell death observed during I/R injury in free flaps. On one hand, apoptosis contributes to localized tissue damage and cellular dysfunction, thereby exacerbating flap ischemia. On the other hand, when only a subset of mitochondria undergo minority MOMP, mitochondrial DNA (mtDNA) is released into the cytosol, and low levels of caspase activity are maintained. This process triggers inflammatory responses and impairs macrophage-mediated clearance, facilitating the secondary necrosis of apoptotic cells and further amplifying tissue damage [7]. Consequently, to enhance flap survival, investigators have employed pharmacological agents and pretreatment strategies—including estradiol (E2), melatonin, platelet-rich plasma (PRP) pretreatment, betulinic acid (BA), caloric restriction (CR), and pravastatin—to upregulate the anti-apoptotic protein Bcl-2 and downregulate pro-apoptotic proteins such as Bax, Caspase-3, and Caspase-8 [21,22,23,24,25].

Furthermore, research indicates that pathways such as JNK/p38 MAPK and NF-κB are closely associated with the occurrence of apoptosis when IRI occurs in flaps. Wallner et al. demonstrated that the p38 MAPK and NF-κB pathways were suppressed in ischemic/reperfused flaps following MSTN knockout in mice, resulting in a significant reduction in apoptosis compared to the control group [40]. Additionally, Wang et al. demonstrated that suppression of the p38 MAPK pathway following Bax gene knockout significantly reduced apoptosis within the skin flaps post-I/R injury [41]. Preconditioning with hyperbaric oxygen (HBO) suppresses apoptosis by reducing the levels of phosphorylated JNK (pJNK) [42]. The selective JNK inhibitor SP600125, and remifentanil, among other pharmacological agents, mitigate I/R-induced apoptosis and improve flap outcomes by downregulating the ASK-1/JNK pathway [43,44].

Several pharmacological agents, such as Ganoderma lucidum polysaccharide peptide (GLPP) and L-borneol, facilitate flap survival by suppressing cell apoptosis through the inhibition of JNK/p38 MAPK and NF-κB pathway activation, thereby downregulating MAPK and NF-κB levels [45,46].

Additionally, it has been demonstrated in several studies that the activation of the PI3K/Akt pathway potently stimulates cellular proliferation while concurrently inhibiting apoptosis within the flap tissues. Employing chemically modified mRNA (modRNA) encoding SDF-1α can activate the SDF-1α/CXCR4 signaling axis, subsequently initiating PI3K/Akt signaling, thereby enhancing angiogenesis and fibroblast proliferation within ischemic flaps while simultaneously suppressing apoptosis [47]. Similarly, pharmacological interventions such as tetramethylpyrazine (TMP), luteolin, biliverdin (Bv), and leonurine (Leo) can also attenuate apoptosis within ischemic flaps via the activation of Akt [48,49,50,51].

### 4.2. Autophagy in Flap Necrosis: A Double-Edged Sword and Therapeutic Modulation Strategies

In the context of cell death, autophagy exhibits a dual role in the context of flap IRI. On one hand, it can clear damaged mitochondria via mitophagy, thereby reducing the generation of ROS and preventing further damage. On the other hand, excessive activation of autophagy leads to the excessive degradation of healthy mitochondria, thereby diminishing cellular survival [52].

It has been demonstrated that the inhibition of excessive autophagy can be achieved through the activation of aldehyde dehydrogenase-2 (ALDH2) by its specific activator Alda-1. This process suppresses excessive mitophagy via the PINK1/Parkin pathway, thereby promoting flap survival following IRI [53]. Furthermore, drugs such as notoginseng triterpenes (NTs) and dimethyl-2-oxoglutarate (DMOG) can also inhibit excessive autophagy through various pathways, thereby promoting flap survival [54,55]. Interestingly, Zhengtai Chen et al. demonstrated that while Sitagliptin enhances flap survival by inhibiting apoptosis and promoting angiogenesis, the induced autophagy is detrimental to flap survival. The precise mechanisms underlying this adverse effect warrant further investigation [56].

### 4.3. Pyroptosis in Flap Necrosis: Inflammatory Pathways and Therapeutic Modulation

Following flap reconstruction, ischemia within the flap induces an elevation in intracellular ROS levels. This, in turn, activates key pyroptosis mediators, including NLRP3, Caspase-1, and GSDMD-N, leading to cellular membrane rupture and the subsequent release of pro-inflammatory cytokines such as IL-1β and IL-18. Consequently, the localized inflammatory response is amplified, thereby accelerating cell death and flap necrosis. Existing studies have demonstrated that inhibition of the NLRP3/caspase-1 axis or direct suppression of GSDMD cleavage can effectively mitigate ischemia/reperfusion injury across multiple organs, including the brain, heart, and liver [57]. Similarly, studies have revealed that the inhibition of NLRP3 inflammasome activation during flap ischemia significantly enhances flap survival. Pharmacological agents demonstrating efficacy include cathelicidin BF-30, cyclic helix B peptide (CHBP), and neuregulin-1 (NRG1) [26,27,39].

Furthermore, activation of NF-κB from Toll-like receptor 4 (TLR4) subsequently promotes NLRP3 inflammasome expression, thereby facilitating pyroptosis [58]. Pharmacological interventions, including paeoniflorin, rivaroxaban, and saxagliptin, have demonstrated the capacity to inhibit this pathway [33,59,60]. This inhibition leads to a reduction in NLRP3 inflammasome expression and a concomitant decrease in pro-inflammatory cytokines, thereby suppressing pyroptosis and inflammation, ultimately enhancing flap survival rates. Li et al. also found that pinocembrin (Pino) activates SIRT3 through the AMPK/PGC-1α pathway, which can reduce oxidative stress and pro-angiogenic production and inhibit pyroptosis and apoptosis to improve flap survival [61].

### 4.4. Necroptosis in Flap Necrosis: Pathophysiology and Targeted Inhibition Strategies

Flap necrosis resulting from necroptosis is often exacerbated by RIP kinase-mediated necroptosis within the ischemic region of the flap. This process intensifies endothelial damage, thereby diminishing flap survival [62]. Consequently, current investigations employ diverse therapeutic strategies to suppress RIP-mediated necroptosis and enhance flap survival. These include the use of RIP pathway inhibitors such as NecroX-5 and Necrostatin-1 [63,64]. These interventions target the Rip1/Rip3/Mlkl pathway, thereby significantly mitigating injury and promoting recovery from IRI, ultimately improving flap survival rates [65].

In addition, Lou et al. demonstrated that Mir504-5p can be utilized to regulate PLA2G4E. Overexpression of Mir504-5p inhibits PLA2G4E and its induced lysosomal membrane permeabilization (LMP), thereby suppressing necroptosis in ischemic flaps and promoting flap survival [66].

### 4.5. Ferroptosis in Flap Necrosis: Iron Overload, Lipid Peroxidation, and Therapeutic Interventions

Ferroptosis is primarily driven by iron accumulation and lipid peroxidation. Excessive iron ions promote lipid peroxidation, leading to the generation of lipid peroxides, such as 4-hydroxynonenal (4-HNE) and malondialdehyde (MDA), in the cell membrane, ultimately causing membrane rupture and cell death [28]. Glutathione peroxidase 4 (GPX4) acts as a critical enzyme in preventing lipid peroxidation during this process. In studies investigating the inhibition of ferroptosis-mediated flap necrosis, GPX4 is also considered as a critical regulator of ferroptosis. The expression level of GPX4 is commonly investigated to assess the efficacy of pharmacological interventions in suppressing ferroptosis. For instance, caloric restriction (CAL), exendin-4, kaempferol (KAM), and maresin 1 (MaR1) have all demonstrated the ability to enhance flap survival by elevating GPX4 levels within the flap tissue, thereby inhibiting ferroptosis [28,35,36,67].

The observed reduction in 4-HNE and MDA levels within free flaps, alongside the upregulation of NRF2 and SLC7A11, further supports the notion of ferroptosis inhibition. Yu et al. demonstrated that apoptotic bodies (ABs) derived from fibroblast-like cells within the subcutaneous connective tissue of the flap can target KEAP1 via miR-339-5p, thereby suppressing the KEAP1/Nrf2 axis. This, in turn, inhibits ferroptosis and promotes the survival of ischemic flaps [34].

### 4.6. PANoptosis in Flap Necrosis: A Multifaceted Cell Death Mechanism and Potential Therapeutic Strategies

Upon tissue IRI, key cell death regulators, including Caspase-8, Caspase-3, RIPK1, RIPK3, and MLKL, are activated, thereby inducing the concurrent occurrence of pyroptosis, apoptosis, and necroptosis, collectively termed PANoptosis. This multifaceted cell death mechanism critically compromises flap survival [68]. Recently, it has been reported that platelet-derived small extracellular vesicles (PL-sEVs) can simultaneously inhibit key proteins associated with apoptosis, pyroptosis, and necroptosis, thereby reducing the activation of the PANoptosis pathway in flaps. Furthermore, PL-sEVs exhibit the capacity to promote angiogenesis, downregulate pro-inflammatory cytokine levels, and suppress the NF-κB pathway, which regulates PANoptosis. The PLEL@PL-sEV hydrogel system facilitates the stable and controlled release of PL-sEVs, thereby significantly enhancing flap survival through sustained, long-term therapeutic effects [31].

## 5. Crosstalk Among PCD-Related Signaling Pathways in Flap Necrosis

Previous studies have identified that in the context of IRI, several types of PCD could concurrently occur. The interaction and crosstalk among these PCDs have led to research on pharmacological intervention that targets upper stream signaling pathways (Figure 3).

Cathelicidin BF-30 activates TFEB via the AMPK–TRPML1–calcineurin pathway, thereby enhancing autophagy and mitigating ROS-induced pyroptosis and apoptosis [26]. CHBP can also activate autophagy by activating the AMPK–TRPML1–calcineurin pathway, leading to TFE3 activation, and subsequently inhibition of necroptosis and pyroptosis in ischemic flaps [27].

The mTOR signaling pathway has been reported to function as a primary negative regulator of autophagy [69]. When mTOR is active, it suppresses autophagy by inhibiting the activity of autophagy-related proteins, such as ULK1. CAL, via the AMPK–mTOR–TFEB signaling pathway, suppresses mTOR and initiates autophagy, thereby inhibiting ferroptosis and apoptosis in ischemic flaps [28]. Exosomes derived from fibroblast growth factor 1 (FGF1)-pretreated adipose-derived stem cells (FEXOs) improved flap survival under ischemic conditions by activating autophagy and inhibiting apoptosis and pyroptosis via the PI3K–Akt–mTOR pathway [29].

Another critical signaling is the canonical NF-κB signaling pathway, which is primarily activated by inflammatory cytokines (TNF-α, IL-1β), PAMPs, ROS, and various stressors (e.g., UV radiation, DNA damage). Its principal functions involve promoting inflammatory responses, inhibiting apoptosis-related proteins, and participating in immune regulation [70]. Consequently, modulating the NF-κB signaling pathway, such as through the PPAR-γ/NF-κB and TLR4–NF-κB, can effectively suppress the inflammatory response and promote flap survival during ischemia. Pharmaceuticals such as Prussian blue nanozyme (PBzyme), PL-sEVs, catalpol, and rivaroxaban suppress the inflammatory response in ischemic flaps via these pathways, concurrently inhibiting multiple modes of cell death, including apoptosis, pyroptosis, necroptosis, and PANoptosis [30,31,32,33].

Nrf2 is a critical component of the body’s antioxidant defense system, activating a cascade of antioxidant and detoxification genes to maintain cellular homeostasis. Under normal conditions, Nrf2 is primarily negatively regulated by KEAP1, influencing cell death by modulating intracellular oxidative stress levels [71]. Yu et al. demonstrated that ABs derived from fibroblast-like cells (FSCTs) promote the survival of ischemic flaps by suppressing ferroptosis and reducing oxidative stress via the miR-339-5p/KEAP1/Nrf2 axis, concurrently inducing M1-to-M2 macrophage polarization [34]. Similarly, MaR1, KAM, and empagliflozin (EMPA) can also inhibit ferroptosis and apoptosis through this pathway, thereby promoting flap survival [35,36,37].

Additionally, the STING (Stimulator of Interferon Genes) signaling pathway may also play a key role in IRI of flaps. STING is a critical regulator of cellular antiviral immune responses, inflammation, and programmed cell death, primarily activated via the cGAS–STING axis [72]. In the context of IRI in flaps, the inhibition of STING activation can attenuate the production of ROS and the activation of the NLRP3 inflammasome, thereby suppressing pyroptosis, apoptosis, and necroptosis. For instance, dihydrocapsaicin (DHC) has been shown to inhibit various forms of cell death via the cGAS–STING pathway, ultimately promoting flap survival [38]. Furthermore, Xuwei Zhu et al. demonstrated that NRG1 can suppress STING activity via the Akt–FOXO3a signaling pathway, consequently reducing pyroptosis and necroptosis, thereby enhancing flap survival under ischemic conditions [39].

## 6. Novel Biomaterials for Enhancing Flap Survival: Therapeutic Applications and Prospects

Recent investigations have explored the application of diverse biomaterials in flaps subjected to IRI, aiming to mitigate PCD and, consequently, enhance flap survival. Hydrogels represent one of the most extensively researched biomaterials in contemporary scientific inquiry. As carriers for targeted therapy across multiple diseases, hydrogels exhibit unique advantages due to their biocompatibility, injectability, self-healing properties, and controlled release capabilities, making it particularly valuable in targeted therapeutic applications [73,74]. Xu et al. developed a composite calcium silicate–human serum albumin (CS-HSA) hydrogel, employing HSA hydrogel as a carrier and cross-linking it with CS to achieve enhanced therapeutic effects. The CS component facilitates the release of silicon ions, which, in turn, activate the Akt/JNK–MAPK pathway while concurrently suppressing the STAT1 pathway. Furthermore, HSA serves as a biocompatible and injectable drug carrier, capitalizing on its inherent advantages. The resultant composite hydrogel significantly improves flap survival rates by promoting angiogenesis and mitigating both cell apoptosis and inflammatory responses [75]. Similarly, Liu et al. integrated platelet-derived small extracellular vesicles (PL-sEVs) with a thermosensitive triblock hydrogel (PLEL), culminating in the creation of a PLEL@PL-sEV spray delivery system. The PL-sEVs not only enabled a stable release and uniform distribution of their contents but also, by leveraging their growth factor-rich properties, promoted angiogenesis and tissue regeneration. Furthermore, they inhibited the NF-κB pathway, thereby mitigating inflammation and suppressing PANoptosis, offering novel molecular targets for flap therapy [31].

Furthermore, Hou et al. pioneered the use of Prussian blue nanozymes (PBzymes) to maintain a stable microenvironment in transplanted flaps. This was achieved by selectively modulating the PPAR-γ/NF-κB and Bax/Bcl-2/Caspase-3 pathways, thereby suppressing inflammatory responses, apoptosis, and necroptosis [30]. Subsequently, researchers have combined decellularized dermal matrix (d-ECM), Prussian blue nanoparticles (PB NPs), and alprostadil (Alp), leveraging their respective biocompatibility, antioxidant, and anti-inflammatory properties, as well as their vasodilatory effects, to inhibit apoptosis and improve flap survival rates [76].

In flap transplantation research, other biomaterials have also been explored. For instance, Wang et al. employed microfluidic 3D printing to create MXene-incorporated hollow fibrous (MX-HF) scaffolds, loaded with VEGF, to achieve customized repair of complex tissue defects. The dynamic response and spatiotemporally controlled drug release of these scaffolds not only effectively inhibited cell apoptosis and inflammatory responses but also enhanced the targeting of the therapy, ultimately leading to a significant improvement in flap survival rates, thus providing an innovative solution for flap regeneration [77]. Furthermore, Lu et al. engineered a microneedle patch encapsulating a cobalt-based metal–organic framework (Co-MOF@MN) to achieve the sustained transdermal release of Co^2^⁺, thereby activating the HIF-1 pathway. This modulation of angiogenesis and inflammation, coupled with the inhibition of apoptosis, mitigates ischemia/reperfusion injury and markedly enhances flap survival [78].

## 7. Limitations and Future Directions

The issues associated with free flap-related PCD are multifaceted. Firstly, the understanding of complex regulatory networks and crosstalk among various forms of PCD remains incomplete. Moreover, most studies exploring therapeutic approaches are predominantly based on animal models, with limited clinical validation, and therapeutic strategies involving novel biomaterials demonstrate relatively low translational efficiency and lack optimized controlled-release systems. These limitations hinder the clinical application of research findings. Therefore, future efforts should focus on enhancing translational research, developing more targeted therapeutic interventions, and optimizing controlled-release systems for biomaterials. Additionally, in-depth investigations into the synergistic mechanisms among different forms of programmed cell death, such as apoptosis and ferroptosis, are essential to establish a more robust theoretical foundation for the prevention and treatment of IRI in skin flaps.

## 8. Conclusions

The findings of this review indicate that treatments targeting one or more types of PCD to prevent flap necrosis from IRI may serve as efficient strategies. Moreover, these strategies may enhance therapeutic outcomes with assistance of various types of biomaterials. There is an urgent need for more clinical trials to provide robust validation.

## Figures and Tables

**Figure 1 biomolecules-15-00911-f001:**
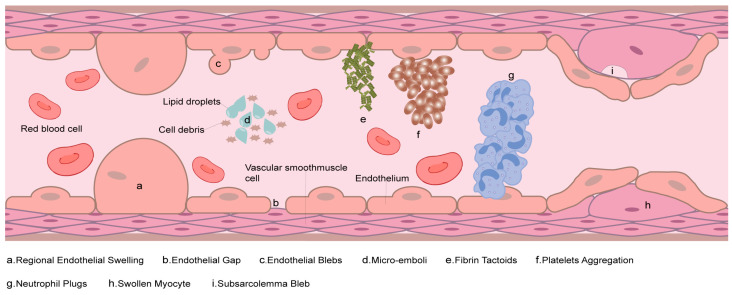
Key pathological changes in microcirculation during IRI. (**a**) Regional endothelial swelling with cytoplasmic edema and luminal narrowing. (**b**) Endothelial gap formation due to tight junction degradation, permitting RBC extravasation. (**c**) Endothelial blebs reflecting membrane protrusions from cytoskeletal disruption. (**d**) Micro-emboli composed of lipid droplets or apoptotic debris obstructing capillaries. (**e**) Fibrin tactoids as spindle-shaped fibrin polymers within thrombi. (**f**) Platelet aggregation with pseudopod extension and microthrombus formation. (**g**) Neutrophil plugs adhering to activated endothelium. (**h**) Swollen myocytes compressing adjacent vessels via osmotic edema. (**i**) Subsarcolemmal blebs beneath the sarcolemma from calpain-mediated cytoskeletal damage.

**Figure 2 biomolecules-15-00911-f002:**
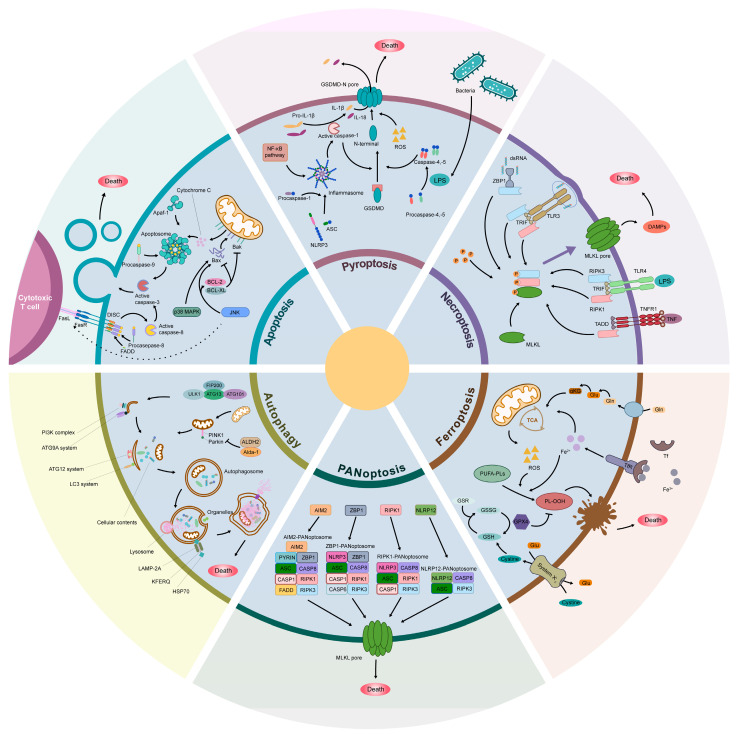
Schematic of six major forms of PCD associated with flap necrosis. Apoptosis, pyroptosis, necroptosis, ferroptosis, PANoptosis, and autophagy are major types of PCD that have been studied in the context of IRI and flap necrosis. Apoptosis is a caspase-dependent programmed cell death process initiated via intrinsic (mitochondrial/Bcl-2 family-mediated cytochrome c release) or extrinsic (death receptor/FasL-TNFR1-Caspase-8) pathways, characterized by cell shrinkage and DNA fragmentation. Pyroptosis is inflammatory cell death triggered by Caspase-1/4/5/11 cleavage of gasdermin D, forming membrane pores and releasing IL-1β/IL-18, often activated by inflammasomes. Necroptosis is a regulated necrosis mediated by RIPK1-RIPK3-MLKL signaling, causing plasma membrane rupture and DAMPs release, typically triggered by Caspase-8 inhibition. Ferroptosis is an iron-dependent lipid peroxidation-driven death caused by GPX4 inactivation or FSP1-CoQ10 axis disruption, linked to redox imbalance and cystine/glutamate antiporter (System Xc⁻) inhibition. PANoptosis is a coordinated inflammatory death pathway integrating pyroptosis, apoptosis, and necroptosis via ZBP1, AIM2, or NLRP12 sensors, engaging caspases, RIPKs, and gasdermins. Autophagy is a lysosome-mediated degradation process regulated by ATG proteins to recycle organelles and nutrients that can promote survival or death depending on context.

**Figure 3 biomolecules-15-00911-f003:**
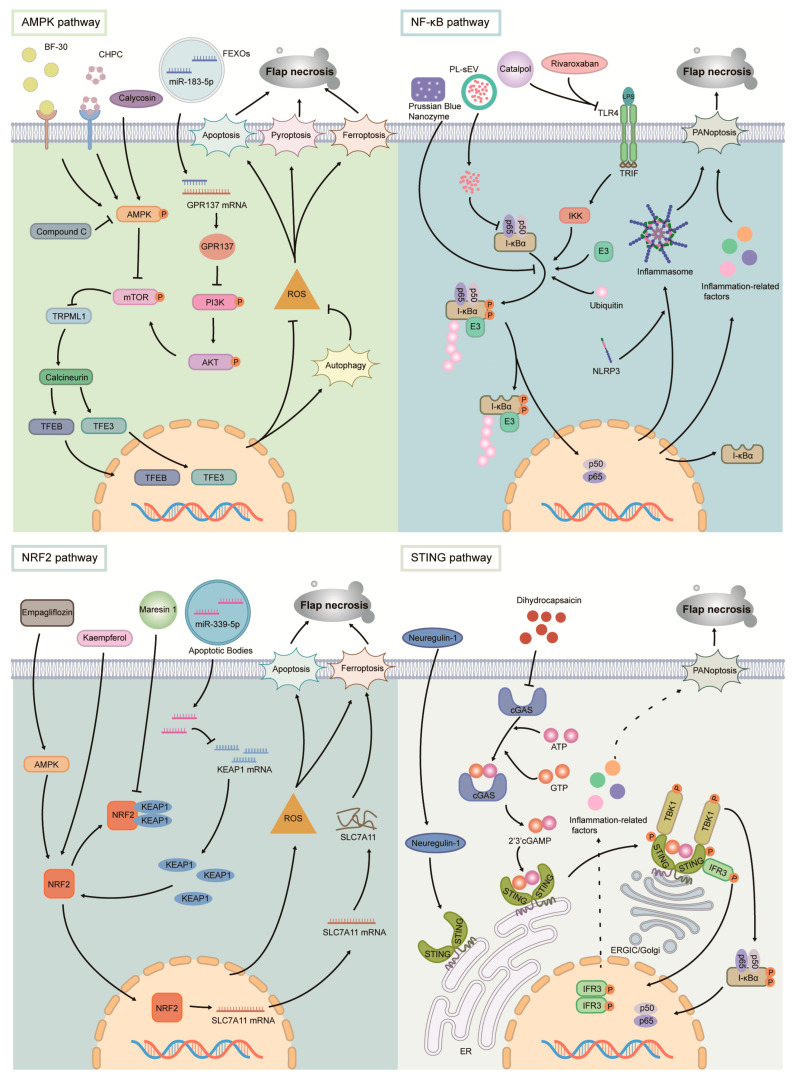
Major signaling pathways regulating multiple types of PCD in flap necrosis. By utilizing different drugs, the activation of the AMPK/NF-κB/NRF2/STING signaling pathways can be modulated to suppress programmed cell death, as well as the production of inflammatory cytokines and reactive oxygen species (ROS), thereby effectively inhibiting flap necrosis.

**Table 1 biomolecules-15-00911-t001:** Summary of drugs targeting PCD to prevent IRI.

Drugs	Time/Termination	Outcome Measures	Target Spot	Target Pathways	PCD	Therapeutic Effect of the Medication Group	Model of Experiment	Ref.
BF-30	Day 7 after the operation	Blood supply, survival area and microvascular density, collagen damage, ROS, GSH, and MDA levels	TFEB	AMPK/TRPML1/calcineurin	Apoptosis↓, Pyroptosis↓, Autophagy↑	No significant change in the necrotic area; blood flow and the number of blood vessels increase, while collagen damage decreases	C57BL/6J mice models with I/R-injured island skin flap	N. Yang et al. (2024) [26]
CHBP	Day 7 after the operation	Blood supply, survival area, tissue edema, collagen damage, microvascular density, ROS, and MDA levels	TFE3	AMPK/TRPML1/calcineurin	Pyroptosis↓, Necroptosis↓, Autophagy↑	Blood flow and the number of blood vessels increase, while the necrotic area and tissue edema decrease	C57BL/6J mice models with random-pattern skin flap	Lou et al. (2022) [27]
CAL	Day 7 after the operation	Blood supply, survival area, microvascular density, tissue edema collagen volume fraction, ROS, SOD, MDA, and GSH levels	SOD1	AMPK/mTOR	Apoptosis↓, Ferroptosis↓, Autophagy↑	Blood flow and the number of blood vessels increase, while the necrotic area, tissue edema and subcutaneous venous congestion decrease	Rat models with modified McFarlane flap model	Jiang et al. (2024) [28]
FEXO	Day 7 after the operation	Blood supply, survival area, the temperature of skin flaps, and ROS levels	GPR137	PI3K/AKT/mTOR	Apoptosis↓, Pyroptosis↓	Blood flow and the number of blood vessels increase, while the necrotic area, the temperature of skin flaps and ROS levels decrease	C57BL/6J mice models with random-pattern skin flap	Zhang et al. (2024) [29]
PBzymes	Day 7 after the operation	Blood supply, survival area, microvascular density, and ROS levels	PPAR-γ, NF-κB, Bax, Bcl-2, caspase-3	PPAR-γ/NF-κB; Bax/Bcl-2/Caspase-3	Apoptosis↓, Necroptosis↓	Blood flow and microvascular density increase, while the necrotic area and ROS level decrease	Mice models with I/R-injured skin flap; HUVEC hypoxia-reoxygenation model.	Hou et al. (2022) [30]
PL-sEV	Day 7 after the operation	Blood supply, survival area, microvascular density, collagen volume fraction, the temperature of skin flaps, and ROS levels	p-IκBα, p-P65	NF-κB	PANoptosis↓	Blood flow, microvascular density, and collagen volume fraction increase, while the necrotic area, temperature of skin flaps and ROS level decrease	C57BL/6J mice models with random-pattern skin flap; oxygen–glucose deprivation/reoxygenation (OGD/R) injury on HUVEC	Liu et al. (2024) [31]
Catalpol	Day 7 after the operation	Blood supply, survival area, microvascular density, SOD, and MDA levels	TLR4, NF-κB	TLR4/NF-κB	Pyroptosis↓	Blood flow, microvascular density, and SOD level increase, while the necrotic area, and MDA level decrease	Rat models with skin flap	Ma et al. (2024) [32]
Rivaroxaban	Day 7 after the operation	Blood supply, survival area and microvascular density	TLR4, NF-κB	TLR4/NF-κB	Pyroptosis↓	Blood flow, microvascular density, and increase, while the necrotic area decreases	Rat models with McFarlane flaps	Wang et al. (2024) [33]
ABs	Day 7 after the operation	Blood supply, survival area, the temperature of skin flaps, collagen content, MDA, and GSH levels	KEAP1	miR-339-5p/KEAP1/Nrf2	Ferroptosis↓	Blood flow, microvascular density collagen content, and GSH level increase, while the necrotic area, the temperature of skin flaps, and MDA level decrease	C57BL/6J mice models with random-pattern skin flap	G. Yu et al. (2024) [34]
MaR1	Day 7 after the operation	Blood supply, survival area, microvascular density, collagen content, ROS, SOD, MDA, and GSH levels	KEAP1, Nrf2	KEAP1/Nrf2	Apoptosis↓, Ferroptosis↓	Blood flow, microvascular density collagen content, SOD, and GSH levels increase, while the necrotic area, ROS, and MDA levels decrease	Rat models with McFarlane flaps	Fang et al. (2024) [35]
KAM	Day 7 after the operation	Blood supply, survival area, SOD, and MDA levels	Nrf2, SIRT1,TNF, NF-κB, Toll-like receptor, and NOD-like receptor	HMGB1/TLR4/NF-κB and Nrf2/SLC7A11/GPX4	Ferroptosis↓	Blood flow and SOD level increase, while the necrotic area and MDA level decrease	Rat models with McFarlane flaps	Wang et al. (2025) [36]
EMPA	Day 7 after the operation	Blood supply, survival area, SOD, and MDA levels	AMPK, Nrf2, GPX4	AMPK/Nrf2/GPX4	Ferroptosis↓	Blood flow and SOD level increase, while the necrotic area and MDA level decrease	Rat models with modified McFarlane flap model	Yang et al. (2025) [37]
DHC	Day 7 after the operation	Blood supply, survival area, collagen damage, microvascular density, ROS, GSH, and MDA levels	NLRP3	c-GAS/STING	PANoptosis↓	Blood flow and microvascular density increase, while the necrotic area, collagen damage, ROS, GSH, and MDA levels decrease	Rat models with deep circumflex iliac artery (DCIA-flap)	Lai et al. (2024) [38]
NRG1	Day 7 after the operation	Blood supply, survival area, microvascular density	AKT, FOXO3a	Cgas/STING	Pyroptosis↓, Necroptosis↓	Blood flow and microvascular density increase, while the necrotic area decreases	C57BL/6 mice models with modified McFarlane flap model	Zhu et al. (2024) [39]

ABs: apoptotic bodies; BF-30: Cathelicid-BF; CAL: calcyphosin; CHBP: Cyclic Helix B Peptide; DHC: dihydrocapsaicin; EMPA: empagliflozin; FEXO: exosomes derived from FGF1-pretreated adipose-derived stem cells; IRI: ischemia/reperfusion injury; KAM: kaempferol; MaR1: maresin 1; NRG1: Neuregulin-1; PBzymes: Prussian blue nanozymes; PCD: programmed cell death; PL-sEVs: platelet-derived small extracellular vesicles.

## Data Availability

Not applicable.

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
