# Peer review of "Targeting Programmed Cell Death in Flap Ischemia/Reperfusion Injury"

_biomolecules, 2025, doi:10.3390/biom15070911_

Round 1
Reviewer 1 Report
Comments and Suggestions for Authors
This review comprehensively summarizes the mechanisms and therapeutic strategies targeting programmed cell death (PCD) in the context of flap ischemia-reperfusion injury (IRI). The article is logically organized and effectively integrates recent advances in PCD subtypes, including ferroptosis and PANoptosis, thereby highlighting both its theoretical underpinnings and translational potential. Nevertheless, several issues remain to be addressed prior to publication, such as redundant conten and inconsistent use of terminology.
Detailed Comments
- Strengths
- Relevance: The focus on PCD in flap IRI addresses a significant and unmet clinical challenge in reconstructive surgery.
- Comprehensive Coverage: The detailed mechanisms of apoptosis, autophagy, pyroptosis, necroptosis, ferroptosis, and PANoptosis are thoroughly explained, with clear support from informative figures (Figs 1-3).
- Data Integration: Table 1 provides an effective summary of drugs targeting PCD along with their corresponding experimental models, serving as a valuable resource for further research.
- Areas for Improvement
- Logical Flow:
Redundancy exists between Sections 3 and 4. Streamline mechanistic descriptions and emphasize therapeutic linkages. - Clinical Translation:
Overreliance on animal models; include clinical trial data (e.g., Necrostatin-1) to bridge preclinical and clinical gaps. - Literature Update:
Some references (e.g., Dragu et al., 2011) are outdated. It is recommended to incorporate updated references, such as: International Journal of Pharmaceutics: X, 2025, 9: 100334; Angewandte Chemie International Edition, 2025, 64: e202505669. - Title
Based on the manuscript content, the following revisions are recommended to improve the logical coherence and clarity of Sections 3-6 titles:
Section 3 Title: The term "Concepts" is redundant, as the section focuses on in-depth analysis of specific signaling pathways rather than abstract definitions. Revise to "Signaling Pathways and Regulatory Networks of Programmed Cell Death" to align with the mechanistic emphasis in the text.
Section 4 Title: The title lacks specificity and fails to highlight the dual focus on mechanisms and therapeutic strategies. Modify to "Mechanisms and Therapeutic Strategies of Programmed Cell Death in Flap Necrosis" to better reflect the section’s content, which systematically discusses PCD subtypes (e.g., pyroptosis, necroptosis) and their targeted interventions.
Section 5 Title: The term "Regulating" inaccurately implies unidirectional control, while the text emphasizes bidirectional interactions between pathways (e.g., AMPK/NF-κB/NRF2/STING crosstalk). Reframe to "Crosstalk Among PCD-Related Signaling Pathways in Flap Necrosis" to emphasize the dynamic interplay described in the manuscript.
Section 6 Title: The title overly narrows the scope to "future directions," while the section details both current applications (e.g., hydrogels, nanozymes) and future prospects of biomaterials. Revise to "Novel Biomaterials for Enhancing Flap Survival: Therapeutic Applications and Prospects" to comprehensively cover the section’s focus on biomaterial-driven therapies and their translational potential.Open with unmet need: “Metastatic cancers remain challenging due to heterogeneity and therapy resistance.”
- Figure/Table Optimization:
Carefully check and ensure consistency in the use of terminology throughout the text. For example, in Table 1 (e.g., unify the format of 'PI3K/Akt/mTOR' versus 'PI3K-Akt-mTOR').
Author Response
Comment 1: This review comprehensively summarizes the mechanisms and therapeutic strategies targeting programmed cell death (PCD) in the context of flap ischemia-reperfusion injury (IRI). The article is logically organized and effectively integrates recent advances in PCD subtypes, including ferroptosis and PANoptosis, thereby highlighting both its theoretical underpinnings and translational potential. Nevertheless, several issues remain to be addressed prior to publication, such as redundant conten and inconsistent use of terminology.
Response 1: We thank the reviewer for the appreciation of our work. Based on the reviewer's comments, we have made the following revisions and additions.
Detailed Comments
Comment 2: 1. Strengths
Relevance: The focus on PCD in flap IRI addresses a significant and unmet clinical challenge in reconstructive surgery.
Comprehensive Coverage: The detailed mechanisms of apoptosis, autophagy, pyroptosis, necroptosis, ferroptosis, and PANoptosis are thoroughly explained, with clear support from informative figures (Figs 1-3).
Data Integration: Table 1 provides an effective summary of drugs targeting PCD along with their corresponding experimental models, serving as a valuable resource for further research.
Response 2: We are grateful for the recognition of our work by the reviewer.
- Areas for Improvement
Comment 3: Logical Flow:
Redundancy exists between Sections 3 and 4. Streamline mechanistic descriptions and emphasize therapeutic linkages.
Response 3: Thank you for pointing out the key issue. We have streamlined the content of the third section and inserted a transitional paragraph between the third and fourth sections to enhance the article's flow and reduce its mechanical nature.
Comment 4: Clinical Translation:
Overreliance on animal models; include clinical trial data (e.g., Necrostatin-1) to bridge preclinical and clinical gaps.
Response 4: Thanks for the suggestions. Unfortunately, we failed to find any clinical trial related to PCD-targeted therapy. However, we believe that in the near future, more clinical translations will emerge, as there is a relatively sufficient amount of evidence from preclinical models.
Comment 5: Literature Update:
Some references (e.g., Dragu et al., 2011) are outdated. It is recommended to incorporate updated references, such as: International Journal of Pharmaceutics: X, 2025, 9: 100334; Angewandte Chemie International Edition, 2025, 64: e202505669.
Response 5: We fully concur with the reviewer's suggestion. We have updated the references, including the two mentioned above, to adhere to MDPI's standard of at least 50% recent literature for review articles.
Comment 6: Title
Based on the manuscript content, the following revisions are recommended to improve the logical coherence and clarity of Sections 3-6 titles:
Section 3 Title: The term "Concepts" is redundant, as the section focuses on in-depth analysis of specific signaling pathways rather than abstract definitions. Revise to "Signaling Pathways and Regulatory Networks of Programmed Cell Death" to align with the mechanistic emphasis in the text.
Section 4 Title: The title lacks specificity and fails to highlight the dual focus on mechanisms and therapeutic strategies. Modify to "Mechanisms and Therapeutic Strategies of Programmed Cell Death in Flap Necrosis" to better reflect the section’s content, which systematically discusses PCD subtypes (e.g., pyroptosis, necroptosis) and their targeted interventions.
Section 5 Title: The term "Regulating" inaccurately implies unidirectional control, while the text emphasizes bidirectional interactions between pathways (e.g., AMPK/NF-κB/NRF2/STING crosstalk). Reframe to "Crosstalk Among PCD-Related Signaling Pathways in Flap Necrosis" to emphasize the dynamic interplay described in the manuscript.
Section 6 Title: The title overly narrows the scope to "future directions," while the section details both current applications (e.g., hydrogels, nanozymes) and future prospects of biomaterials. Revise to "Novel Biomaterials for Enhancing Flap Survival: Therapeutic Applications and Prospects" to comprehensively cover the section’s focus on biomaterial-driven therapies and their translational potential.Open with unmet need: “Metastatic cancers remain challenging due to heterogeneity and therapy resistance.”
Response 6: We thank the reviewer for the valuable suggestions on titles. We totally agree with the reviewer and have revised the titles of sections 3-6. However, the last sentence of the suggestion appears to be irrelevant to our manuscript.
Comment 7: Figure/Table Optimization:
Carefully check and ensure consistency in the use of terminology throughout the text. For example, in Table 1 (e.g., unify the format of 'PI3K/Akt/mTOR' versus 'PI3K-Akt-mTOR').
Response 7: We have verified the consistency in the use of terminology throughout the text and corrected the error in Table 1.
Once again, we thank the reviewer for all the helpful advice. We have made modifications based on these seven comments in the manuscript. We believe that these changes have significantly improved the logicality, reliability, and readability of this review, which now meets the acceptance criteria of MDPI.
Reviewer 2 Report
Comments and Suggestions for Authors
General comments
The manuscript entitled “Targeting Programmed Cell Death in Flap Ischemia-Reperfusion Injury” addresses a timely and important topic in reconstructive surgery and regenerative medicine. The authors present a thorough review of the various forms of programmed cell death (PCD) implicated in ischemia-reperfusion injury (IRI) of skin flaps, along with associated therapeutic strategies. The topic is relevant and well chosen, and the review captures the complex interplay between cell death mechanisms and tissue viability.
The manuscript demonstrates a commendable effort to synthesize preclinical data and highlight molecular pathways with therapeutic relevance. Particularly valuable is the discussion on novel biomaterials and combinatory approaches that modulate multiple PCD pathways. However, in its current form, the manuscript would benefit from several major revisions to strengthen its scientific rigor, bibliographic validity, and translational relevance.
Specific comments
- Abstract
The abstract is clear and informative, offering a good overview of the topic. That said, the final sentence feels overly general and could be made more concise and precise. Consider rephrasing to better reflect the specific value of the findings.
- Introduction
The introduction successfully frames the clinical challenge of flap necrosis and introduces PCD as a relevant therapeutic target. However, it could be strengthened by elaborating on why current literature insufficiently addresses the crosstalk between PCD modalities in this context. A clearer rationale for the review focus would increase its impact.
- Methodology
Although a search strategy is mentioned, the section lacks detail. It would be helpful to specify inclusion/exclusion criteria, the review method used (e.g., narrative vs. systematic), and whether any quality assessment of the included studies was performed. The absence of reference to PRISMA or similar frameworks slightly reduces the transparency of the review process.
- Results and thematic synthesis
The article is well organized, with each PCD modality discussed in dedicated sections. The mechanisms are accurately described, and the molecular targets are linked with therapeutic outcomes in relevant animal models. However, this review is heavily reliant on preclinical murine studies, with limited discussion on translational hurdles or clinical evidence. Several sections are overloaded with biochemical detail that could be summarized more concisely to improve readability. Importantly, only approximately 35–40% of the references cited are from the past five years, which falls below MDPI’s recommended standard of at least 50% recent literature for review articles.
Many references stem from the same research groups or institutions in East Asia. While this reflects valuable work, a broader geographic representation would enhance objectivity and reduce potential regional citation bias.
- Figures and Tables
The figures are informative and well designed. The table summarizing therapeutic agents and their experimental models is particularly valuable. I would suggest the addition of a comparative diagram that maps the PCD types against targeted therapies and outcomes, which could help readers better grasp the translational implications.
- Discussion and conclusions
The discussion provides a useful synthesis of findings, but the conclusions remain somewhat generic. The authors should more clearly articulate the current gaps in knowledge (e.g., lack of long-term functional outcomes, limited clinical studies). They should add a critical appraisal of preclinical model limitations and clear directions for future research, particularly emphasizing clinical validation and biomaterial translation.
- References and Literature Balance
Out of over 120 cited works, fewer than half were published within the past five years. It is advisable to incorporate at least 10–15 additional recent sources (from 2022–2024), ideally from high-impact journals and with a wider international scope. This will help to strengthen the article’s relevance and contemporaneity.
Recommendation: Major Revisions
While the manuscript presents significant value and strong scientific grounding, it requires revisions to align with MDPI’s editorial standards and to optimize its scientific contribution. Improving the methodological clarity, updating and diversifying the bibliography, and critically refining the discussion section would substantially strengthen the manuscript and prepare it for eventual acceptance.
Author Response
General comments
Comment 1: The manuscript entitled “Targeting Programmed Cell Death in Flap Ischemia-Reperfusion Injury” addresses a timely and important topic in reconstructive surgery and regenerative medicine. The authors present a thorough review of the various forms of programmed cell death (PCD) implicated in ischemia-reperfusion injury (IRI) of skin flaps, along with associated therapeutic strategies. The topic is relevant and well chosen, and the review captures the complex interplay between cell death mechanisms and tissue viability.
The manuscript demonstrates a commendable effort to synthesize preclinical data and highlight molecular pathways with therapeutic relevance. Particularly valuable is the discussion on novel biomaterials and combinatory approaches that modulate multiple PCD pathways. However, in its current form, the manuscript would benefit from several major revisions to strengthen its scientific rigor, bibliographic validity, and translational relevance.
Response 1: We thank the reviewer for the appreciation of our work. Based on the reviewer's comments, we have made the following revisions and additions.
Specific comments
Abstract
Comment 2: The abstract is clear and informative, offering a good overview of the topic. That said, the final sentence feels overly general and could be made more concise and precise. Consider rephrasing to better reflect the specific value of the findings.
Response 2: Thank you for the suggestion. We had rephrased the last sentence of Abstract to make it more concise and precise. The specific value of the findings in this review is to summarize the mechanisms of PCD in the context of IRI, thereby inspiring novel strategies in future research, which has been mentioned in the Abstract. Due to the word count restriction, it may not be suitable to clarify in the last sentence.
Introduction
Comment 3: The introduction successfully frames the clinical challenge of flap necrosis and introduces PCD as a relevant therapeutic target. However, it could be strengthened by elaborating on why current literature insufficiently addresses the crosstalk between PCD modalities in this context. A clearer rationale for the review focus would increase its impact.
Response 3: We totally agree with the reviewer. We have make a supplementary explanation of the current inadequacy of the crosstalk among different types of PCD in Introduction.
Methodology
Comment 4: Although a search strategy is mentioned, the section lacks detail. It would be helpful to specify inclusion/exclusion criteria, the review method used (e.g., narrative vs. systematic), and whether any quality assessment of the included studies was performed. The absence of reference to PRISMA or similar frameworks slightly reduces the transparency of the review process.
Response 4: We appreciate for the valuable suggestions from the reviewer. This review is a narrative review, which may not necessarily require the quality assessment of the included studies or the PRISMA framework. However, we totally agreed that methods should be detailedly exhibited. Therefore we make modifications to Methodology, including the exclusion criteria.
Results and thematic synthesis
Comment 5: The article is well organized, with each PCD modality discussed in dedicated sections. The mechanisms are accurately described, and the molecular targets are linked with therapeutic outcomes in relevant animal models. However, this review is heavily reliant on preclinical murine studies, with limited discussion on translational hurdles or clinical evidence. Several sections are overloaded with biochemical detail that could be summarized more concisely to improve readability. Importantly, only approximately 35–40% of the references cited are from the past five years, which falls below MDPI’s recommended standard of at least 50% recent literature for review articles.
Response 5: We thank the reviewer for pointing out the inadequacy. Actually, current studies are primarily within the range of preclinical models, particularly those involving mice. We believed that this may indicate that the research field is still in the early stages, but with strong translational potential. Therefore, we added an extra section "7. Limitations and Future Directions" to discuss the hurdles and the need for clinical translations. Furthermore, we streamlined the content of the third part to make it more coherent and easier to read. Most importantly, we have updated the references to comply with MDPI's standard of at least 50% recent literature for review articles.
Comment 6: Many references stem from the same research groups or institutions in East Asia. While this reflects valuable work, a broader geographic representation would enhance objectivity and reduce potential regional citation bias.
Response 6: Thank you for the suggestion. All studies related to PCD in flap necrosis have been thoroughly screened, and the references included in this review meet the inclusion criteria mentioned in the Methodology section. However, we are also surprised to find that major research institutions are located in East Asia, while other regions have also participated in the research field. We believe that this may indicate that these organizations in East Asia are the most active participants, and may provide novel preclinical evidence for PCD-targeting therapies.
Figures and Tables
Comment 7: The figures are informative and well designed. The table summarizing therapeutic agents and their experimental models is particularly valuable. I would suggest the addition of a comparative diagram that maps the PCD types against targeted therapies and outcomes, which could help readers better grasp the translational implications.
Response 7: Thank you for your valuable advice. We completely agree with you that the outcomes of PCD-targeted therapies should be highlighted. However, adding a diagram for this part may overlap with the context in Figure 2. Therefore, we have incorporated this information into Table 1 and made modifications to Table 1 to help readers better understand the translational implications.
Discussion and conclusions
Comment 8: The discussion provides a useful synthesis of findings, but the conclusions remain somewhat generic. The authors should more clearly articulate the current gaps in knowledge (e.g., lack of long-term functional outcomes, limited clinical studies). They should add a critical appraisal of preclinical model limitations and clear directions for future research, particularly emphasizing clinical validation and biomaterial translation.
Response 8: We thank the reviewer for the suggestion. As previously mentioned, we have included an additional section titled "7. Limitations and Future Directions" to critically appraise the limitations of preclinical models and to provide clear directions for future research, with a particular emphasis on clinical validation and biomaterial translation.
References and Literature Balance
Comment 9: Out of over 120 cited works, fewer than half were published within the past five years. It is advisable to incorporate at least 10–15 additional recent sources (from 2022–2024), ideally from high-impact journals and with a wider international scope. This will help to strengthen the article’s relevance and contemporaneity.
Response 9: Thanks for the advice. As mentioned above, we have updated the references to comply with MDPI's standard of at least 50% recent literature for review articles.
Recommendation: Major Revisions
Comment 10: While the manuscript presents significant value and strong scientific grounding, it requires revisions to align with MDPI’s editorial standards and to optimize its scientific contribution. Improving the methodological clarity, updating and diversifying the bibliography, and critically refining the discussion section would substantially strengthen the manuscript and prepare it for eventual acceptance.
Response 10: Once again, we sincerely thank the reviewer for the valuable suggestions. We have made modifications based on these ten comments in the manuscript. We believe that these changes have significantly improved the logicality, reliability, and readability of this review, which now meets the acceptance criteria of MDPI.
Round 2
Reviewer 1 Report
Comments and Suggestions for Authors
After the author's revisions, the quality of the manuscript has significantly improved. However, there is a minor error in the format of Ref 74, and please change "n/a" to "64".
Author Response
Comment: After the author's revisions, the quality of the manuscript has significantly improved. However, there is a minor error in the format of Ref 74, and please change "n/a" to "64".
Response: We have revised the manuscript based on the reviewer's opinion. Thanks again to the reviewer for your appreciation for our work.
Reviewer 2 Report
Comments and Suggestions for Authors
The manuscript could be accepted for publication in this revised form.
Author Response
Comment: The manuscript could be accepted for publication in this revised form.
Response: Thank you very much to the reviewer for taking the time to review this manuscript and thank you again for your appreciation of our work.